# Preliminary Assessment of Four Wild Leafy Species to Be Used as Baby Salads

Ada Baldi *, Stefania Truschi, Piero Bruschi and Anna Lenzi

Department of Agriculture, Food, Environment and Forestry (DAGRI), University of Florence Piazzale delle Cascine 18, 50144 Firenze, Italy; stefania.truschi@unifi.it (S.T.); piero.bruschi@unifi.it (P.B.); anna.lenzi@unifi.it (A.L.)
* Correspondence: ada.baldi@unifi.it

**Abstract:** Wild edible leafy plants, thanks to their organoleptic characteristics and nutritional value that can make them be appreciated as salads by consumers, represent a good opportunity for growers and the fresh-cut industry, which are always looking for new crops to expand the number of products they offer. In this study, four wild species (dandelion, sorrel, wild chicory, and wild lettuce) were cultivated hydroponically up to the baby leaf stage in order to evaluate them as potential crops. At harvest, yield and antioxidant compounds, minerals, and nitrates content were assessed. The contribution to human mineral intake and the possible health risk associated with heavy metals were investigated. A characterization of the sensory profile was also carried out. Yield and chlorophylls and carotenoids content of the investigated species were comparable to those of common leafy vegetables. Variability in nitrate content was observed, with the lowest value in sorrel and the highest in dandelion. All species could contribute in Cr, Mg, and Se intake, and health risks due to heavy metals were excluded. Each species was well characterized by distinctive and peculiar sensory notes. In conclusion, the results of this preliminary study suggest that the four wild investigated species may be promising for baby leaf production.

**Keywords:** *Taraxacum campylodes* G.E.Haglund; *Rumex acetosa* L.; *Cichorium intybus* L.; *Lactuca serriola* L.; leafy vegetables; yield and quality; nitrate; dietary intake; health risk; sensory profile

## 1. Introduction

The term "baby leaf" refers to the young leaves of vegetable crops harvested up to the eighth true leaf and are mainly consumed as salads, both singularly and in multi-species mixes [1]. Although possibly commercialized as unprocessed products, baby leaves are usually minimally processed as ready-to-eat salads, a market globally valued at USD 10.78 billion in 2020 and expected to expand further in the coming years [2]. The worldwide success of this food category is to be found in the fact that it combines excellent health properties due to the high content of nutrients and antioxidant compounds, coupled with fresh consumption, with the advantage of ease of use; thus, it meets both the demand of consumers increasingly aware of the health benefits of a diet rich in fresh vegetables and the need for quick food preparation typical of modern life [3]. For the fresh-cut industry, baby leaves have the advantage of undergoing a lower degree of cutting than adult vegetables, which makes them less subject to browning of the surfaces of the cut edges and to the release of nutrients favorable to the growth of bacteria [4].

A high number of species belonging to different botanical families are grown as baby leaves, of which lettuce (*Lactuca sativa* L.) (Asteraceae), with many types, is the most important, followed by rocket (*Eruca vesicaria* ([L.] Cav.) and wild rocket (*Diplotaxis tenuifolia* ([L.] DC.) (Brassicaceae) [1,3]. The large assortment of baby leaf crops provides a wide range of shapes, colours, tastes, and textures. Nevertheless, consumers are constantly demanding diversification of products, and consequently, growers and the fresh-cut industry are always

looking for new crops to expand the number of goods they offer. Ethnobotany may be a source of inspiration for this purpose, as wild edible flora is rich in leafy species that are gathered and consumed as salad [5]. In Mediterranean countries, wild greens have always been an important part of the daily diet [6]. For example, in the Tuscany region (Italy), among 357 taxa of wild food plants traditionally used in the local gastronomy, 220 are herbaceous plants utilized as leafy vegetables, and, of 17 different recipe groups, salads are the second largest category [5]. Some of these species (e.g., *Cichorium intybus* L.) are progenitors of crops but could further be exploited for agricultural purposes, as during domestication, the strong selective pressure could have led to the loss of precious alleles and characters [7]. Some others (e.g., *Taraxacum campylodes* G.E.Haglund) have been subjected to domestic or semi-amateur cultivation experiences (very small areas, local markets) but deserve to be taken into consideration for cultivation on a larger scale. Finally, other interesting wild edible plants are still completely uncultivated [8]. For example, among Asteraceae, *Helminthotheca echioides* (L.) Holub and *Hypochoeris radicata* L., present in the list of the first 30 most cited wild edible species in Tuscany according to the ethnobotanical study of Baldi et al. [5], have never been cultivated.

In general, a number of factors make wild food plants promising and convenient candidates as new crops. Because they are usually more tolerant than crop plants to adverse climatic and edaphic conditions [9,10] and to pests and diseases [11], they can lead to agricultural systems which are more sustainable and resilient to climate change [12]. From a chemical point of view, a high content of vitamins and minerals is typical of wild greens [13–15]. In some cases, the amount even exceeds that found in cultivated vegetables or other food, which are considered typical sources of that specific nutrient. For example, *Bunias erucago* L. has an iron content much higher than spinach and meat [16]. Moreover, a wide variety of phytochemicals with antioxidant effects have been reported in many of these species [17], and some contain molecules showing antimicrobial potential [18] or other biological-pharmacological activities [19].

In the particular case of baby leaves, the typical early growth stage would make the introduction of wild species into cultivation, in theory, relatively simpler than for crops that have to reach successive and more complex developmental stages, such as the induction of flowering, the fruit setting, the development of underground organs, the formation of the head, etc., thus requiring complicated domestication processes. The safety issue is also to be considered. The cultivation of wild plants would reduce some health risks that the consumption of specimens collected in the wild can entail. These risks are due to the possible accumulation of pollutants such as nitrates and heavy metals, often present in high quantities in the soils in which they live [20,21]. Finally, it is important to remember that wild plants used for food represent a bio-cultural heritage that agriculture could help to safeguard from the risks of progressive depletion linked to the ongoing disappearance of the rural society [5].

The characterization of potential candidates for agricultural exploitation may be the first step for the introduction of wild species into cultivation. The aim of this research was to characterize four common wild edible leafy species (*T. campylodes*, *Rumex acetosa* L., *C. Intybus*, and *Lactuca serriola* L.) from an agronomic, chemical, nutritional, safety, and sensory point of view in order to preliminary evaluate their suitability as baby leaf crops. The first three were chosen for their popularity, and because they had already been grown as minor species [22], *L. serriola* was selected as the ancestor of lettuce, the most important baby leaf crop. In addition, a criterion of choice was the availability of the seeds.

## 2. Materials and Methods

### 2.1. Plant Material, Growing Conditions, and Data Collection

Plants of dandelion (*Taraxacum campylodes* G.E.Haglund), sorrel (*Rumex acetosa* L.), wild chicory (*Cichorium intybus* L.), and wild lettuce (*Lactuca serriola* L.) were hydroponically grown in a floating system up to the baby leaf stage (Figure 1).

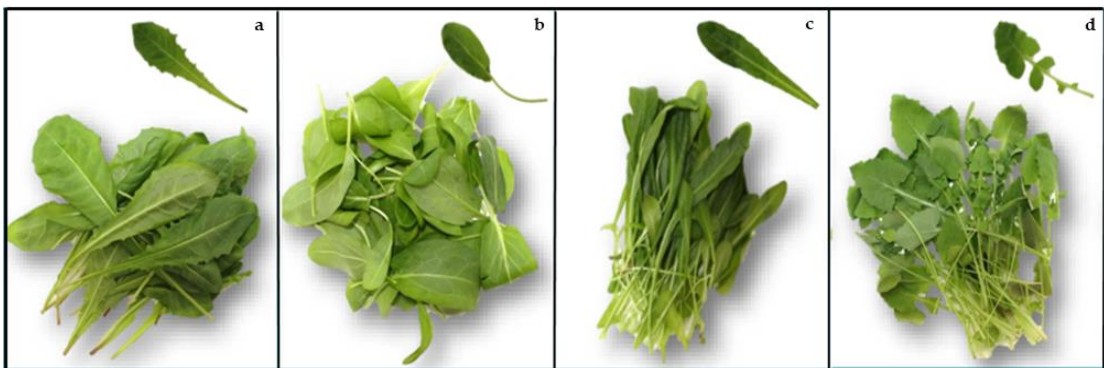

**Figure 1.** The four edible wild species investigated in the study: (**a**) dandelion, (**b**) sorrel, (**c**) wild chicory, and (**d**) wild lettuce.

Seeds used as starting material were purchased from "B & T World seeds" (Aiguesvives, France), Provencemonamour (Paris, France), and Fratelli Ingegnoli (Milan, Italy), respectively, for sorrel, wild chicory, and wild lettuce, while seeds of dandelion were collected in the wild in an uncultivated peri-urban area of Lucca (Central Italy) in spring 2021.

Seeds were sown in polystyrene cell trays (L 227.5 mm × W 130.8 mm; 28 cells, Ø 27 mm) filled with vermiculite (Perlite Italiana srl, Corsico, Milan, Italy). Four seeds per cell were sown for each species (3840 seed $m^2$). After sowing, trays were placed in the dark in a germination chamber at 20 °C for 48 h. Afterward, trays were put in polyethylene terephthalate tanks (L 260.0 mm × W 180.0 mm × H 80 mm) containing 1.8 L of standard Hoagland's nutrient solution [23] prepared with distilled water (macronutrients in mM and micronutrients in μM: 15 N, 1 P, 6 K, 5 Ca, 2 Mg, 50 Fe, 46.2 B, 9.2 Mn, 0.78 Zn, 0.32 Cu, 0.12 Mo; pH 5.52, electric conductivity (EC) 1.1 mS/cm) and moved into a walkin growth chamber. Here, plants were grown at 24 ± 2 °C (day) and 17 ± 2 °C (night) with a photoperiod of 16 h under fluorescent lighting units OSRAM L36 W/77 (Osram, Munich, Germany). Once a week, tanks were refilled with fresh nutrient solution up to the initial volume.

Plants were harvested 7 weeks after sowing. One part of the harvested material was used to carry out the sensory profile characterization, while another part was used to determine the following parameters: number of plants; fresh weight (FW); dry weight (DW) after oven-drying at 50 °C until a constant weight; leaf area (LA), measured by an area meter LI-3100 (LI-COR, Lincoln, NE, USA); SPAD index, measured immediately before harvest on three fully expanded leaves randomly selected from each tank by means of SPAD-502 chlorophyll meter (Konica-Minolta, Tokyo, Japan); leaf colour, measured immediately after harvest by means of an NR-3000 Portable Colorimeter (Nippon Denshoku Kogyo C., LTD., Tokyo, Japan) according to the CIE system and expressed as L* (lightness or darkness), a* (redness or greenness) and b* (yellowness or blueness) values; and contents of minerals and metals (Al, Ba, Ca, Cd, Cr, Cu, Fe, K, Mg, Mn, Mo, Na, Ni, P, Pb, Se, Sr and Zn), chlorophylls, carotenoids, and nitrate.

*2.2. Chemical Analysis*

2.2.1. Elemental Composition

Elemental composition was quantified by using 0.5 mg of milled dry sample digested with 10 mL of $HNO_3$ (67% *v/v*) in Teflon reaction vessel and then mineralized in a microwave oven (Mars 5, CEM Corp., Matthews, NC, USA) using the program 1600 W, 100% power, at 200 °C for 20 min. At the end of mineralization, the final volume of 25 mL was reached by adding ultra-pure water. The concentrations of Al, Ba, Ca, Cd, Cr, Cu, Fe, K, Mg, Mn, Mo, Na, Ni, P, Pb, Se, Sr, and Zn were determined using an inductively coupled argon plasma optical emission spectrometer (ICP–OES iCAP series 7000 Plus Thermo Scientific, Waltham, MA, USA). A standard method for the 18 different elements was applied, using the Qtegra[TM] Intelligent Scientific Data Solution[TM] (ISDS), and the wavelengths selected

were 394.4 nm for Al, 493.4 nm for Ba, 315.9 nm for Ca, 228.8 for Cd, 267.7 for Cr, 324.8 nm for Cu, 259.9 nm for Fe, 766.5 nm for K, 285.2 nm for Mg, 257.6 nm for Mn, 204.6 nm for Mo, 589.6 nm for Na, 231.6 nm for Ni, 178.8 nm for P, 220.4 nm for Pb, 196.1 nm for Se, 421.6 nm for Sr, and 206.2 nm for Zn quantification. The calibration was performed with several dilutions of the multi-element standard Astasol®-Mix (ANALYTIKA®, spol. s.r.o., Prague, Czech Republic) in 1% $HNO_3$ ($v/v$) at different concentrations (0.1, 1, 10 and 100 mg/L) with the exception of P, for which the calibration was performed at a P concentration of 10 and 100 mg/L. Blanks and appropriate certified reference materials ERM®-CD281 (B-2440 Geel, Belgium) were included in each batch digested for quality control. Measurements were performed in triplicate. Data were expressed on an FW basis considering the fresh weight/dry weight ratio.

### 2.2.2. Chlorophyll and Carotenoids

Chlorophylls and carotenoids were extracted from fresh tissues (about 50 mg) using methanol 99.9% as solvent. Samples were kept in a dark room at 4 °C for 24 h, and the assays were carried out immediately after extraction. Chlorophyll *a* (Chl *a*) and chlorophyll *b* (Chl *b*) were determined by the increase in absorbance at 665.2 nm and 652.4 nm, respectively. Total chlorophyll content (Chl *ab*) was determined as the sum of Chl *a* and Chl *b*. Carotenoid content was computed by the increase in absorbance at 470 nm. The pigment concentrations were calculated by Lichtenthaler's formula [24].

### 2.2.3. Nitrate

Total nitrate content was determined by spectrophotometer using the salicylic acid method [25]. The analysis was performed on samples of 100 mg of dried powdered leaf tissue (80 °C for 48 h) suspended in 10 mL of deionized water on an orbital shaker at room temperature for 2 h. Subsequently, 70 µL of aqueous extract was mixed with 300 µL of 5% salicylic acid in sulphuric acid and with 10 mL of 1.5 M NaOH. The solution was cooled at room temperature for 20 min before reading the absorbance at 410 nm, and the nitrate concentration was calculated through a $KNO_3$ standard calibration curve. Data were expressed on an FW basis considering the fresh weight/dry weight ratio.

### 2.3. Contribution to Mineral Requirement and Health Risk Assessment

For evaluating how much the studied species can contribute to human mineral requirements, the amount of mineral elements potentially taken through the consumption of the different baby leaves was calculated as Estimated Dietary Intake (EDI, mg/day) using the following formula:

$$EDI = C_{mineral} \times (CP/1000) \tag{1}$$

where $C_{mineral}$ is the element concentration (mg/kg FW) in the produce and CP is the consumed portion of baby leaves per day per person, which was assumed to be 50 g [26]. Then, EDI was expressed as percentage (EDI%) of the recommended dietary intake (RDI, mg/day) (for Ca, Cu, Fe, K, Mg, Mo, Na, P, Se, and Zn) or adequate intake (AI, mg/day) (for Cr and Mn) as defined by Italian Society of Human Nutrition (SINU), considering RDI and AI values referred to an adult male [27].

In order to assess the possible health risk due to the intake of heavy metals related to consumption of the baby leaves, the health risk index (HRI) was calculated for the metals detected in leaves of the investigated species according to the following formula:

$$HRI = EDI_{Bw}/RfD \tag{2}$$

where $EDI_{Bw}$ is the EDI (as defined above) per kg of body weight (BW) and RfD (mg/kg BW/day) is the oral reference dose, which is an estimate of the daily exposure of humans to heavy metals having no hazardous effect during the lifetime according to US-EPA [28]. Since the US-EPA database currently lacks an RfD for Al and Cu, the possible health risk was evaluated on the basis of the tolerable weekly intake (TWI; mg/kg BW/week) of

Al reported by EFSA [29] and Cu RfD according to Taylor et al., 2023 [30]. For BW, an average body weight for an adult was considered and assumed to be 70 kg, as in previous studies [31].

### 2.4. Sensory Evaluation

The sensory evaluation was conducted by applying the Consensus Profile method as described in ISO 13299:2016 Standard [32] and ISO 5492:2008 Standard [33] at the Mérieux NutriSciences Lab (Prato, Italy).

A group of three trained assessors with broad experience in the sensory evaluation of leafy vegetables were selected to elicit sensory attributes characterizing the products and to assign them the intensity on a 1–5 quantitative scale. Ten attributes were elicited and divided into four sensory modalities: appearance (green colour); odour (herbaceous); flavour (sour, sweet, bitter, and herbaceous aroma); and texture (crunchiness, chewiness, chilliness, and astringency). The selected attributes, their definition, and the intensity scale are shown in Table 1.

**Table 1.** Attributes, their definitions, and intensity scale used to characterize baby leaves of dandelion, sorrel, wild chicory, and wild lettuce.

| Attribute | Attribute Definition | Intensity Scale |
|---|---|---|
| Green colour | Assessment of the green colour tone of the products. | 1 = light green; 5 = dark green |
| Herbaceous odour | Intensity of the odour attributable to the herbaceous and green notes perceived directly by the olfactory system. | 1 = absent; 5 = high |
| Sour | Basic taste typical of organic acid (i.e., citric acid) perceptible inside the oral cavity. | 1 = absent; 5 = high |
| Sweet | Basic taste typical of sugar (i.e., sucrose) perceptible inside the oral cavity. | 1 = absent; 5 = high |
| Bitter | Basic taste typical of caffeine and quinine perceptible inside the oral cavity. | 1 = absent; 5 = high |
| Herbaceous aroma | Intensity of the flavour attributable to herbaceous and green notes perceived indirectly by the olfactory system. | 1 = absent; 5 = high |
| Crunchiness | Property linked to the modality of deformation of the product and to the intensity of the characteristic sound generated during the breaking phase. The product breaks and reproduces the characteristic sound. | 1 = low; 5 = high |
| Chewiness | Characteristic that measures the deformation capacity of the product following its compression and evaluates the ability of the product to return to its original shape without breaking. | 1 = low; 5 = high |
| Chilliness | Burning sensation perceived in the throat or diffusely in the oral cavity. | 1 = absent; 5 = high |
| Astringency | A sensation characterized by contraction of the gums, increase of dryness and roughness on the tongue, and marked decrease in salivation, which commonly occurs by eating unripe fruits. | 1 = absent; 5 = high |

Immediately before the sensory evaluation session, products were washed and dried by centrifuge. Then, an anonymous sample (identified by a three-digit number) of 100 g for each species was put in a saucer and randomly administered to each assessor. Assessors individually evaluated one sample at a time, recording the intensity of each sensory attribute. At the end of the sensory evaluation session of a single product, the results were collected by the panel leader, which led to a general discussion to reach a mutually agreed consensus for the profile definition. This procedure was repeated until all products were evaluated. Products were evaluated in one replicate. Statistical analysis is not required for this method.

### 2.5. Experimental Design and Statistical Analysis

Twelve replicates per species (1 replicate = 1 tank) were arranged in a completely randomized block design. Data were expressed as mean $\pm$ standard deviation and subjected to one-way analysis of variance (ANOVA). Significant differences between means ($n = 3$–9) were calculated by the post hoc Tukey's test at $p < 0.05$ using CoStat Software (Version 6.45, Monterey, CA, USA).

## 3. Results

### 3.1. Crop Production and Quality

A low emergence was observed in all the species tested in this study. At harvest, seedling emergence percentage was 13.4%, 34.2%, 36.6%, and 43.0% for dandelion, wild lettuce, sorrel, and wild chicory, respectively. Differences in yield were noticed between the species, with sorrel and wild chicory showing higher yields than dandelion and wild lettuce (Figure 2a). On the other hand, dandelion showed higher FW per plant than wild lettuce (Figure 2b), the largest LA (Figure 2c), and a higher DW% than sorrel (Figure 2d).

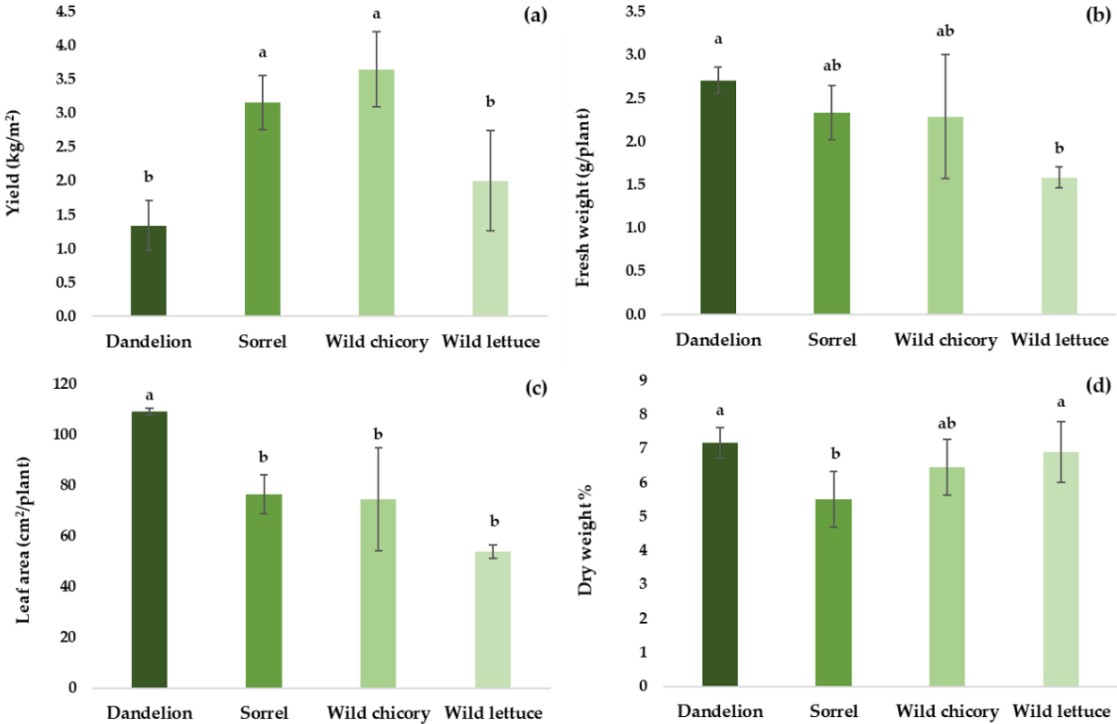

**Figure 2.** Yield (**a**), fresh weight (**b**) and leaf area per plant (**c**), and DW% (**d**) of baby leaves of the four investigated species (means ± standard deviation). Different letters indicate significant differences according to the post hoc Tukey test ($p < 0.05$).

Wild chicory exhibited higher SPAD values than the other species (Table 2). For what concerns colour parameters, wild lettuce was characterized by the lowest saturation of green (a*), higher saturation of yellow (b*) than sorrel, and lower brightness (L*) in comparison with dandelion and wild chicory (Table 2). The highest contents in Chl *a* and *b* and carotenoids were found in wild chicory, while the lowest was in sorrel (Figure 3a–c). Total chlorophyll content followed the same trend of Chl *a* and Chl *b* (data not shown). Sorrel also showed the lowest nitrate value (926 mg/kg FW), which was 3.2, 4.5, and 5.6 times lower than that of wild lettuce (3005 mg/kg FW), wild chicory (4193 mg/kg FW), and dandelion (5227 mg/kg FW), respectively (Figure 3d).

**Table 2.** SPAD values and colour parameters of baby leaves of the four investigated species (means ± standard deviation).

|  | Dandelion | Sorrel | Wild Chicory | Wild Lettuce |
|---|---|---|---|---|
| SPAD | 35.55 ± 2.69 b | 33.01 ± 4.17 b | 40.10 ± 4.03 a | 35.17 ± 3.50 b |
| Colour parameters |  |  |  |  |
| a* | −7.98 ± 0.56 b | −8.74 ± 0.67 b | −7.74 ± 0.55 b | −3.40 ± 1.31 a |
| b* | 26.73 ± 0.49 ab | 23.03 ± 0.69 b | 27.26 ± 3.70 ab | 29.61 ± 2.69 a |
| L* | 42.23 ± 0.90 a | 40.80 ± 2.27 ab | 41.74 ± 1.80 a | 38.03 ± 1.03 b |

Different letters in the same column indicate significant differences ($p < 0.05$) according to the post hoc Tukey test.

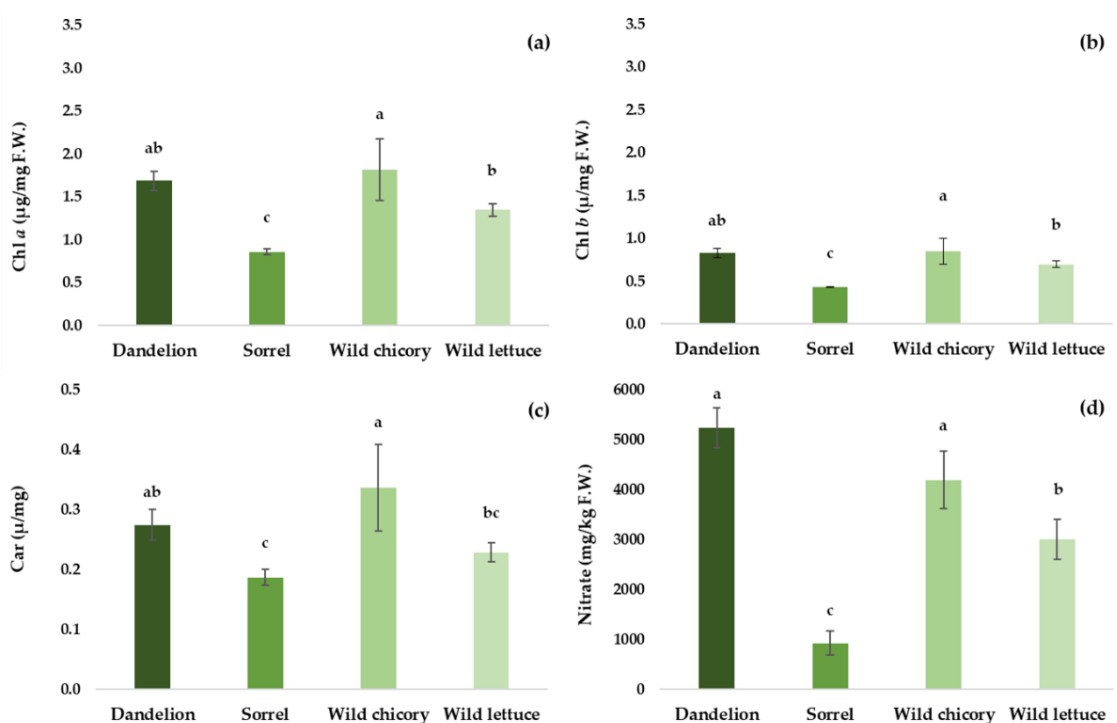

**Figure 3.** Chlorophylls (Chl *a* and Chl *b*) (**a**,**b**), carotenoids (Car) (**c**), and nitrate content (**d** in baby leaves of the four investigated species (means ± standard deviation). Different letters indicate significant differences according to the post hoc Tukey test ($p < 0.05$).

Lead and Cd were not detected by ICP analysis, and no difference in Al, Ba, Cr, Fe, Mn, Ni, P, Se, and Zn content was found. On the contrary, the investigated species showed significant differences in Ca, Cu, K, Mg, Mo, Na, and Sr concentrations (Table 3). Dandelion was the richest in Cu, Mo, and Sr and contained a higher amount of Ca and K than sorrel. Sorrel showed a greater concentration of Mg than dandelion and wild lettuce. The maximum amount of Na was detected in wild lettuce and the minimum in sorrel.

**Table 3.** Minerals and metals content of baby leaves of dandelion, sorrel, wild chicory, and wild lettuce (means ± standard deviation).

| Minerals and Metals (mg/kg F.W.) | Dandelion | Sorrel | Wild Chicory | Wild Lettuce |
|---|---|---|---|---|
| Al | 9.22 ± 5.14 b | 9.45 ± 6.46 a | 2.26 ± 0.68 a | 3.13 ± 2.03 a |
| Ba | 0.78 ± 0.21 a | 1.16 ± 0.34 a | 0.72 ± 0.05 a | 0.79 ± 0.33 a |
| Ca | 590.81 ± 42.48 a | 225.07 ± 18.46 b | 467.01 ± 99.29 a | 524.46 ± 49.31 a |
| Cr | 0.14 ± 0.09 a | 0.21 ± 0.08 a | 0.47 ± 0.21 a | 0.26 ± 0.16 a |
| Cu | 1.03 ± 0.06 a | 0.40 ± 0.04 c | 0.44 ± 0.04 bc | 0.61 ± 0.11 b |
| Fe | 16.92 ± 6.16 a | 9.74 ± 4.03 a | 9.15 ± 2.27 a | 8.30 ± 1.19 a |
| K | 3330.59 ± 95.22 a | 2714.08 ± 86.52 b | 3294.62 ± 163.31 a | 3554.13 ± 271.03 a |
| Mg | 426.74 ± 46.08 b | 565.19 ± 17.60 a | 474.31 ± 53.44 ab | 422.03 ± 8.97 b |
| Mn | 2.78 ± 0.25 a | 5.30 ± 0.14 a | 4.93 ± 0.86 a | 5.69 ± 2.74 a |
| Mo | 0.15 ± 0.06 a | 0.03 ± 0.01 b | 0.04 ± 0.01 b | 0.02 ± 0.00 b |
| Na | 107.77 ± 13.94 bc | 39.52 ± 3.31 c | 254.91 ± 11.42 ab | 321.90 ± 134.27 a |
| Ni | 0.04 ± 0.02 a | 0.10 ± 0.05 a | 0.22 ± 0.11 a | 0.09 ± 0.10 a |
| P | 409.56 ± 29.12 a | 459.39 ± 40.68 a | 306.09 ± 23.87 a | 480.54 ± 145.93 a |
| Se | 0.18 ± 0.01 a | 0.10 ± 0.02 a | 0.16 ± 0.05 a | 0.12 ± 0.02 a |
| Sr | 2.24 ± 0.18 a | 0.38 ± 0.08 c | 0.71 ± 0.04 bc | 1.140.51 b |
| Zn | 2.56 ± 0.45 a | 2.55 ± 0.51 a | 1.76 ± 0.57 a | 2.61 ± 0.84 a |

Different letters in the same row indicate significant differences ($p < 0.05$) according to the post hoc Tukey test.

### 3.2. Contribution to Mineral Dietary Intake and Health Risk Assessment

The EDI% values resulting from consuming a portion of 50 g a day of baby leaves of dandelion, sorrel, wild chicory, and wild lettuce are listed in Table 4.

**Table 4.** Estimated dietary intake expressed as a percentage (EDI%) of the recommended dietary intake (RDI) or adequate intake (AI) resulting from the consumption (50 g per day) of baby leaves of dandelion, sorrel, wild chicory, and wild lettuce.

| Mineral | RDI/*AI* [a] (mg day$^{-1}$) | Dandelion | Sorrel | Wild Chicory | Wild Lettuce |
|---|---|---|---|---|---|
| Ca | **1000** | 2.95 | 1.13 | 2.34 | 2.62 |
| Cr | *0.035* | 19.35 | 29.49 | 66.54 | 37.28 |
| Cu | **0.9** | 5.71 | 2.23 | 2.42 | 3.38 |
| Fe | **10** | 8.46 | 4.87 | 4.57 | 4.15 |
| K | **3900** | 4.27 | 3.48 | 4.22 | 4.56 |
| Mg | **240** | 8.89 | 11.77 | 9.88 | 8.79 |
| Mn | *2.7* | 5.15 | 9.81 | 9.13 | 10.53 |
| Mo | **0.045** | 11.34 | 2.39 | 3.04 | 1.41 |
| Na | **1500** | 0.36 | 0.13 | 0.85 | 1.07 |
| P | **700** | 2.93 | 3.28 | 2.19 | 3.43 |
| Se | **0.055** | 16.70 | 9.44 | 14.13 | 11.32 |
| Zn | **11** | 1.07 | 1.06 | 0.73 | 1.09 |

[a] RDI (bold) and AI (italic), according to SINU (2014).

The highest contribution to RDI/AI was reached for Cr by wild chicory, while the lowest for Na by sorrel. However, differences in the contribution to human mineral requirements between the studied species were not remarkable. Considering the average values, the four investigated species contributed to RDI/AI as follows: 0.6% Na, 1.0% Zn, 2.3% Ca, 3.0% P, 3.4% Cu, 4.1% K, 4.5% Mo, 5.5% Fe, 8.7% Mn, 9.8% Mg, 12.9% Se, and 38.2% Cr.

Table 5 shows the $EDI_{bw}$ and HRI of metals. All the $EDI_{bw}$ values were below the recommended RfD, and HRI was far lower than 1. As regards Al, the weekly consumption of 50 g of product per day would not lead to exceeding the TWI limit of 1 mg/kg body weight/week. The calculated values of weekly consumption were 0.05 mg/kg body weight/week for dandelion and sorrel, 0.01 mg/kg body weight/week for wild chicory, and 0.02 mg/kg body weight/week for wild lettuce.

**Table 5.** Estimated daily intake per kg of body weight ($EDI_{BW}$, mg/kg body weight [1] day$^{-1}$) and health risk index (HRI) resulting from the consumption (50 g per day) of baby leaves of dandelion, sorrel, wild chicory, and wild lettuce.

| Metal | | Dandelion | Sorrel | Wild Chicory | Wild Lettuce |
|---|---|---|---|---|---|
| Ba (RfD [2] = 0.2) | $EDI_{BW}$ | 0.000561 | 0.000827 | 0.000517 | 0.000566 |
| | HRI | 0.002803 | 0.004135 | 0.002586 | 0.002828 |
| Cr (RfD = 0.003) | $EDI_{BW}$ | 0.000097 | 0.000147 | 0.000333 | 0.000186 |
| | HRI | 0.032254 | 0.049154 | 0.110908 | 0.062130 |
| Cu (RfD = 0.04) | $EDI_{BW}$ | 0.000734 | 0.000287 | 0.000312 | 0.000435 |
| | HRI | 0.018343 | 0.007165 | 0.007794 | 0.010864 |
| Fe (RfD = 0.7) | $EDI_{BW}$ | 0.012085 | 0.006957 | 0.006533 | 0.005930 |
| | HRI | 0.017265 | 0.009938 | 0.009334 | 0.008471 |
| Mn (RfD = 0.14) | $EDI_{BW}$ | 0.001986 | 0.003784 | 0.003523 | 0.004062 |
| | HRI | 0.014186 | 0.027027 | 0.025165 | 0.029018 |
| Mo (RfD = 0.005) | $EDI_{BW}$ | 0.000105 | 0.000022 | 0.000028 | 0.000013 |
| | HRI | 0.021066 | 0.004437 | 0.005652 | 0.002622 |
| Ni (RfD = 0.02) | $EDI_{BW}$ | 0.000027 | 0.000073 | 0.000156 | 0.000061 |
| | HRI | 0.001357 | 0.003668 | 0.007776 | 0.003074 |

**Table 5.** *Cont.*

| Metal | | Dandelion | Sorrel | Wild Chicory | Wild Lettuce |
|---|---|---|---|---|---|
| Se (RfD = 0.005) | $EDI_{BW}$ | 0.000131 | 0.000074 | 0.000111 | 0.000089 |
| | HRI | 0.026247 | 0.014832 | 0.022199 | 0.017788 |
| Sr (RfD = 0.6) | $EDI_{BW}$ | 0.001599 | 0.000268 | 0.000507 | 0.000817 |
| | HRI | 0.002665 | 0.000447 | 0.000844 | 0.001362 |
| Zn (RfD = 0.3) | $EDI_{BW}$ | 0.001599 | 0.001821 | 0.001258 | 0.001862 |
| | HRI | 0.006096 | 0.006071 | 0.004192 | 0.006207 |

[1] Body weight = 70 kg; [2] RfD = oral reference dose (mg kg$^{-1}$ body weight day$^{-1}$) according to Barnes and Dourson, 1988 [28].

### 3.3. Sensory Evaluation

The intensity of the 10 attributes elicited by assessors for describing the products is listed in Table 6.

**Table 6.** Score assigned to each attribute elicited to characterize baby leaves of dandelion, sorrel, wild chicory, and wild lettuce.

| Attribute | Dandelion | Sorrel | Wild Chicory | Wild Lettuce |
|---|---|---|---|---|
| Green colour | 4.5 | 4.5 | 4.0 | 4.0 |
| Herbaceous odour | 4.0 | 2.5 | 4.0 | 4.5 |
| Sour | 1.0 | 4.5 | 1.0 | 1.0 |
| Sweet | 1.0 | 1.5 | 1.0 | 3.0 |
| Bitter | 4.5 | 2.0 | 5.0 | 3.0 |
| Herbaceous aroma | 4.5 | 2.5 | 4.5 | 5.0 |
| Crunchiness | 4.0 | 4.5 | 4.0 | 4.0 |
| Chewiness | 3.0 | 2.0 | 3.0 | 3.0 |
| Chilliness | 2.0 | 1.0 | 3.0 | 2.0 |
| Astringency | 3.0 | 3.0 | 2.5 | 2.0 |

The sensory attributes were perceived at different intensities in the different species, contributing to the definition of peculiar sensory profiles. Dandelion leaves had a very intense green colour and an intense herbaceous odour. Bitter and herbaceous aromas were the most perceived attributes. A medium–high crunchiness and medium chewiness were perceived by the trained judges. Chilliness was mild, while astringency was perceived at medium intensity. The leaves of sorrel had a very intense green colour with an herbaceous odour perceived at medium–low intensity. Sour, perceived at high intensity, was the prevalent flavour, while bitter and sweet were barely perceptible. As noted for the odour, even regarding aroma, the herbaceous note resulted in medium–low intensity. Sorrel showed high crunchiness and low chewiness. Astringency was perceived at medium intensity. Wild chicory showed intense green-coloured leaves with an intense herbaceous odour. Leaves of wild chicory were very crunchy and showed a medium intensity of chewability. In the mouth, the bitterness was perceived at the highest intensity, followed by an herbaceous aroma. Chilliness and astringency were evaluated as moderately intense. Wild lettuce has intense green-coloured leaves and a very intense herbaceous odour. The herbaceous aroma was intense, while sweetness and bitterness were perceived to be balanced at medium intensity. From a textural point of view, the leaves of wild lettuce were very crunchy and chewable. Astringency and chilly sensation were both evaluated as mild. In particular, sorrel was the only species in which a sour sensation was perceived, and chilliness was absent. No sweet taste was detected in wild chicory and dandelion.

## 4. Discussion

There is very little information in the literature about the growth, yield, and quality at the baby leaf stage of the wild species investigated in this study. The higher yield obtained in sorrel (3.2 kg FW/m$^2$) and wild chicory (3.7 kg FW/m$^2$) compared to dandelion

(1.3 kg FW/m$^2$) was attributable to the greater number of emerged seedlings being the weight of the single plants similar in the three species. The particularly low emergence of dandelion may be due to unfavorable conditions during seed formation or a loss in germination capacity during conservation, considering that the seeds were collected in the wild two years before the cultivation experiment. In a germination test with dandelion seeds pretreated with 2.2% hypochlorite, Lenzi et al. [34] found a 72% germinability. Using these seeds, the authors obtained a yield of about 3.0 kg/m$^2$. A yield (1.0 kg/m$^2$) comparable to that found in this work was achieved by Alexopoulos et al. [35]. Wild lettuce, although having an emergence close to that of sorrel and wild chicory, did not show a difference in yield (2.0 kg FW m$^2$) in comparison with dandelion due to smaller-sized plants. Sorrel, wild chicory, and wild lettuce showed FW per plant and leaf area similar to those observed by Truschi et al. [36] for the same species. Furthermore, our results were consistent with those found by many authors for baby leaf crops grown in soilless culture [37–39]. It is known that high DW% at harvest increases shelf-life in leafy vegetables [40]. In our study, DW% ranged from about 5.0% in sorrel to 7.0% in dandelion. In ten cultivated species at the baby leaf stage, Colonna et al. [41] found values from 5.3% in spinach to 9.4% in rocket.

Colour is an important food quality attribute and plays a significant role in consumers' perception, acceptance, and choice of products [42]. Moreover, many studies report a significant relationship of chromaticity parameters with chlorophyll and total nitrate concentration [43–46]. These authors suggest the use of SPAD meter and colorimeter as non-destructive but accurate analytical methods to estimate chlorophyll and nitrate content. Comparing wild chicory to wild lettuce, our findings were in accordance with these statements. In fact, the results of the chemical analysis revealed higher concentrations of chlorophyll and nitrates in wild chicory than in wild lettuce, consistently with SPAD and the a* and L* values of the two species.

Chlorophylls and carotenoids are an important key factor in crop productivity and quality as they positively influence photosynthetic capacity, visual appearance, and nutraceutical value due to their antioxidant properties [47]. The wild species investigated in this study showed concentrations in chlorophylls and carotenoids close to those observed in leafy vegetables, which are known to be an excellent source of these compounds [34,48–52].

Leafy vegetables may also provide high amounts of minerals in the diet, representing a useful tool to improve human nutritional quality and health status [53]. No information about the elemental composition of wild lettuce baby leaf is available in the literature, but if compared to cultivated lettuce (*L. sativa* L.) grown at the baby leaf stage [41], wild lettuce resulted in being richer in P, Ca, and Mg. The mineral concentration we detected in dandelion, sorrel, and wild chicory was different from that observed by other authors for these species, both collected in nature and cultivated [34,54,55]. Compared to us, these authors found higher concentrations in Ca, Cu, Fe, K, Mn, Na, and Zn but lower in Mg. However, it is worth considering that the mineral content of wild plants may vary significantly depending upon various factors such as genotype, pedoclimatic conditions, season of collection, and developmental stage [5]. The mineral composition is also influenced by the degree of domestication. Ceccanti et al. [54] assessed the composition of some wild leafy species, including sorrel and chicory, observing significant differences between plants gathered in the wild or cultivated. For instance, soilless cultivation of both sorrel and wild chicory resulted in lower and higher content in Ca and Mg, respectively, in comparison with wild collected plants. On the contrary, Disciglio et al. [55] did not observe differences between collected or cultivated wild chicory for Ca and Mg, but Na was higher in cultivated plants.

In accordance with Regulation (EU) No. 1169/2011, foods can be considered significant sources of mineral elements if they contain at least 3000 mg/kg K, 1200 mg/kg Ca, 563 mg/kg Mg, 1050 mg/kg P, 21 mg/kg Fe, 1.5 mg/kg Cu, 15.0 mg/kg Zn, 3.0 mg/kg Mn, 0.06 mg/kg Cr, 0.083 mg/kg Se, and 0.075 mg/kg Mo. Comparing these amounts with those observed in our study, it is possible to state that the investigated species should be a good source of microelements, especially Cr and Se. In fact, EDI% for these elements, considering a daily consumption of a 50 g portion, reached interesting values, ranging from

about 19% (dandelion) to 66% (wild chicory) and from 9% (sorrel) to 17% (dandelion), respectively. Chromium and Se, also called oligoelements or trace elements because they are needed in very low quantities, are of fundamental importance to human health. Chromium facilitates the transport of glucose from the blood to the cells [56], while Se enhances the proper function of the immune system and is linked to cardiovascular diseases and cancer prevention [57]. Considering the macroelements, higher contribution to RDI concerned Mg (9.8 EDI% as the average of the four species), whose content in sorrel reached the reference value for significant sources of minerals according to Regulation (EU) No. 1169/2011.

One of the great concerns related to the consumption of leafy vegetables is their capacity to accumulate nitrates and heavy metals in soil. By passing from the soil to the edible organs, these compounds enter the food chain causing potential human health risks. Nitrates have harmful effects because of their ability to form carcinogenic nitrosamines [58]. The maximum nitrate content allowed for the commercialization of leafy vegetables is set within a broad range (2000–7000 mg $NO_3$/kg) depending on species, season of harvest, and cultivation system by the Commission Regulation (EC) No. 1258/2011. High variability in nitrate content between species was observed in this study. Sorrel showed a nitrate content very far below the lowest recommended limit; wild lettuce and wild chicory fall within the permissible limit set for lettuce, while dandelion exceeded it. In adult plants of dandelion cultivated in a floating system, Alexopoulos et al. [35] and Alexopoulos et al. [59] found a nitrate content ranging from about 600 mg $NO_3$/kg to 4400 mg $NO_3$/kg depending on the pH and EC of the nutrient solution. In Lenzi et al. [34], the nitrate concentration of dandelion baby leaf exceeded 7000 mg/kg FW.

Heavy metals are recognized as a serious threat to human health due to the risk associated with their toxicity to the human body and its proper functions [60]. Accumulation of heavy metals in plants, whether wild or cultivated, depends on several factors, such as metal concentrations in the growth medium and water, metal bioavailability, and environmental conditions [61,62]. Different heavy metals were detected in our baby leaves, including some not directly added to the nutrient solution. Considering that the nutrient solution was prepared with distilled water, it can be hypothesised that these elements were present in trace amounts as contaminants of the fertilizers or of the substrate used for the cultivation [63,64]. Dandelion, sorrel, wild chicory, and wild lettuce are reported in the literature as heavy metal indicators or as hyperaccumulator species [65–68], and this ability could explain the presence of these elements in plant tissues. Nevertheless, the HRI calculated for Ba, Cr, Cu, Fe, Mn, Mo, Ni, Se, Sr, and Zn, considering consumption of 50 g per day of baby leaf, were significantly far lower than 1. Moreover, the calculated weekly consumption of Al did not exceed the TWI recommended by EFSA. Lead and Cd, which are considered highly toxic metals even at very low concentrations, were not detected in the baby leaves. Based on these results, we can conclude that the consumption of the four wild investigated species does not pose health risks in relation to heavy metals, at least in the growing conditions we adopted.

The baby leaves of dandelion, sorrel, wild chicory, and wild lettuce showed a distinctive and peculiar sensory profile, well characterized by specific notes. The basic flavour, the herbaceous note, and the spicy sensation were the sensory areas that most contributed to differentiate them. The bitterness was the dominant attribute of flavour in both dandelion and wild chicory, while sorrel's flavour was characterized by sourness. A balance between sweetness and bitterness was detected in wild lettuce. The crunchiness, which is a relevant feature in salads, was perceived in all the species with high intensity.

## 5. Conclusions

Dandelion, sorrel, wild chicory, and wild lettuce were found to be promising for baby salad production. Their main features can be related to the yield and antioxidant compound content that were comparable to that of leafy vegetables and to the high contribution of Cr, Se, and Mg to dietary intake. From a health point of view, no health risks due to heavy metal accumulation were observed, and sorrel showed a nitrate content very far below the lowest



recommended limit for salads. Furthermore, the sensory evaluation highlighted peculiar differences between species. In dandelion and wild chicory, bitterness was perceived as the dominant flavour, while a sweet taste was absent. Sorrel was the only species in which a sour sensation was sensed, and no chilliness was detected. Wild lettuce showed a balance between sweetness and bitterness. Given these distinctive sensory notes, the investigated species could not only be marketed individually but also as a salad mix in order to meet the consumers' demand for new products. Further research on wild species to be used as baby salads should be encouraged because, given the increasing global population and the growing demand for healthy food, finding alternative human food sources is essential.

**Author Contributions:** Conceptualization, A.B. and A.L.; methodology, A.B., P.B. and A.L.; software, A.B.; validation, A.B. and A.L.; formal analysis, A.B. and A.L.; investigation, A.B., S.T. and A.L.; data curation, A.B., S.T. and A.L.; writing—original draft preparation, A.B. and A.L.; writing—review and editing, A.B., P.B. and A.L.; supervision, A.B., P.B. and A.L. All authors have read and agreed to the published version of the manuscript.

**Funding:** This research received no external funding.

**Data Availability Statement:** The data presented in this study are available on request from the corresponding authors.

**Acknowledgments:** The authors are thankful to Luisa Andrenelli for carrying out the ICP analysis.

**Conflicts of Interest:** The authors declare no conflict of interest.

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
