# Peer review of "Preliminary Assessment of Four Wild Leafy Species to Be Used as Baby Salads"

_horticulturae, doi:10.3390/horticulturae9060650_

Round 1

Reviewer 1 Report

Note Fig.1  (Line 89). Please write Figure 1.

Note Fig.1  (Line 109). Please write Figure 1 in bold font. 

Chapter 2.5 Experimental design and Statistical analysis (Lines 211–215). In this chapter you mentionated oyly ANOVA, but in Table 2,  Figure 2 and 3 you also estimated mean and standard deviation. Please insert this information in Chapter 2.5 Experimental design and Statistical analysis.

Figure 2 and Figure 3. Do error bars denote standard deviations ? Please inset this information in the titles of these figures. Can you change comma to dot in the Y axis of these figures ?  Also please bold notes Figure 2 and Figure 3.

Lines 222, 223. Please change note Fig.2 to Figure 2.

Lines 234–238.  Please change note Fig.3 to Figure 3.

Note Table 1 (line 240). Please bold this note.

Note Table 5 (line 305). Please bold this note.  

Note Table 6 (line 330). Please bold this note.

Author Response

We would like to thank the Reviewer for their constructive and positive feedbacks which helped us to improve the revised manuscript.

We have replied point-by-point to each comment. Line numbers refer to the revised version.

Line 96: Note Fig.1. Please write Figure 1: Done.

Line 116: Note Fig.1 Please write Figure 1 in bold font: Done.

Line 229: Chapter 2.5 Experimental design and Statistical analysis (Lines 211–215). In this chapter you mentionated only ANOVA, but in Table 2, Figure 2 and 3 you also estimated mean and standard deviation. Please insert this information in Chapter 2.5 Experimental design and Statistical analysis: Done.

Line 245 (Figure 2), Line 259 (Table 2), and Line 272 (Figure 3): Do error bars denote standard deviations? Please inset this information in the titles of these figures: Done.

Line 242 (Figure 2) and Line 269 (Figure 3): Can you change comma to dot in the Y axis of these figures?: Done.

Line 244 (Figure 2) and Line 271 (Figure 3): please bold notes: Done.

Lines 244-246: Please change note Fig.2 to Figure 2: Done.

Lines 271-273: Please change note Fig.3 to Figure 3: Done.

Line 214: Note Table 1. Please bold this note: Done.

Line 308: Note Table 5. Please bold this note: Done.

Line 337: Note Table 6. Please bold this note: Done.

Reviewer 2 Report

The manuscript Preliminary evaluation of four wild leafy species to be used as baby salads assess four common wild edible leafy species that can be used as an alternative in human nutrition. The cultivars were described in terms of agronomy, chemistry, nutrition, safety, and sensory perception. The findings are consistent with the study's goals. I congratulate the authors on their work and encourage them to carry out more research in this area and on other species because, given the increasing global population, finding alternatives to human food sources is essential.

Line 2: I suggest switching the word evaluation to assessment in the title. The rationale behind the selection of the four researched species must be included in the introduction section. Why did the authors choose to concentrate on them out of the vast array of plants that are available for consumption during the phase of young leaves, as they also stated at the start of the second paragraph? Lines 56-57: "other interesting wild edible plants are still completely uncultivated". Like which others? I recommend the authors to provide some examples. Many wild plants have phytotoxic alkaloids that keep grazing animals and pests from eating their leaves. What can the authors tell us about the alkaloid content of the studied species, in the context in which these substances can present toxicity for human health? If this aspect was not followed, is this kind of information available in the specialized literature? Lines 82-84: Please rephrase the objective of the research: "In this paper...... were cultivated...." is not a correct expression. The paper presents only the results of the research... The results of the research are well presented through figures and tables that accurately reflect the main findings. I recommend the authors to elaborate a little more on the conclusion section, emphasizing the most important findings of the research.

Author Response

We would like to thank the Reviewer for their constructive and positive feedbacks which helped us to improve the revised manuscript.

We have replied point-by-point to each comment. Line numbers refer to the revised version.

Line 2: I suggest switching the word evaluation to assessment in the title: Done.

Lines 88-91: The rationale behind the selection of the four researched species must be included in the introduction section. Why did the authors choose to concentrate on them out of the vast array of plants that are available for consumption during the phase of young leaves, as they also stated at the start of the second paragraph?: We have reply to this question in the last sentences of the introduction section.

Lines 57-60: "other interesting wild edible plants are still completely uncultivated". Like which others? I recommend the authors to provide some examples: Done.

Many wild plants have phytotoxic alkaloids that keep grazing animals and pests from eating their leaves. What can the authors tell us about the alkaloid content of the studied species, in the context in which these substances can present toxicity for human health? If this aspect was not followed, is this kind of information available in the specialized literature?: We previously carried out a research on the wild edible plants containing toxic alkaloids (Baldi et al., 2022) and the four species evaluated in the present study were not among them. For this reason, we did not consider this topic in our paper.

Baldi, A.; Bruschi, P.; Campeggi, S.; Egea, T.; Rivera, D.; Obón, C.; Lenzi, A. The renaissance of wild food plants: Insights from Tuscany (Italy). Foods 2022, 11(3), 300. https://doi.org/10.3390/foods11030300

Lines 85-88: Please rephrase the objective of the research: "In this paper...... were cultivated...." is not a correct expression: Done.

I recommend the authors to elaborate a little more on the conclusion section, emphasizing the most important findings of the research: Done.

Reviewer 3 Report

The MS "Preliminary evaluation of four wild leafy species to be used as baby salads" is well writen and stuctured. It covers all imprtant aspects of baby leaf product quality - nutritional value and consumer safety. 

Introduction is very comprehensive.

Materials and methods - Line 115: Why temperature 50 C was chosen for oven-drying of plant material? It is rather unusual for dry weight determination.

Formatting: lines 241-247 are shifted left

English: Line 18: There should be "were" instead of "was".

Discussion: Please, provide any explanations to the low germination of dandelion. What are experiences of other authors? Lower yield is the result of lower germination. What would be the yield at comparable with other species germination rate?

Table 5 for sensory evaluation can be transferred to the Materials and Methods.

Probably it is worth mentioning that one of the plants recommended for food at baby leaf stage is poisonous at adult stage.

Line 18: There should be "were" instead of "was"

Author Response

We would like to thank the Reviewer for their constructive and positive feedbacks which helped us to improve the revised manuscript.

We have replied point-by-point to each comment. Line numbers refer to the revised version.

Line 122: Materials and methods: Why temperature 50 °C was chosen for oven-drying of plant material? It is rather unusual for dry weight determination: We dried the samples at 50 °C because this is considered as a thermal threshold to avoid the potential loss of Selenium by volatilization during dehydration.

Lines 260-265: Formatting: lines are shifted left: Done.

Line 18: English:There should be "were" instead of "was": Done.

Lines 345-351: Discussion: Please, provide any explanations to the low germination of dandelion. What are experiences of other authors? Lower yield is the result of lower germination. What would be the yield at comparable with other species germination rate?: Done.

Line 208-216: Table 5 for sensory evaluation can be transferred to the Materials and Methods: Done.

Probably it is worth mentioning that one of the plants recommended for food at baby leaf stage is poisonous at adult stage: We assume that the reviewer refers to Lactuca serriola. Based on the information provided in the TPPT database (plant section), Lactuca serriola is reported as weakly toxic as well as cultivated lettuce (Lactuca sativa) so we preferred not to deal with this topic.

https://www.agroscope.admin.ch/dam/agroscope/de/dokumente/publikationen/tppt-xls.xlsx.download.xlsx/TPPT_database.xlsx